# Cardiotoxicity and Chemotherapy—The Role of Precision Medicine

**DOI:** 10.3390/diseases9040090

**Published:** 2021-12-08

**Authors:** Thyla Viswanathan, Chim C. Lang, Russell D. Petty, Mark A. Baxter

**Affiliations:** 1Dundee School of Medicine, Ninewells Hospital, University of Dundee, Dundee DD2 1SY, UK; t.viswanathan@dundee.ac.uk; 2Division of Molecular and Clinical Medicine, School of Medicine, University of Dundee, Dundee DD2 1SY, UK; c.c.lang@dundee.ac.uk (C.C.L.); r.petty@dundee.ac.uk (R.D.P.); 3UKM Medical Molecular Biology Institute (UMBI), Jalan Yaacob Latif, Cheras, Kuala Lumpur 56000, Malaysia; 4Tayside Cancer Centre, Ninewells Hospital and Medical School, NHS Tayside, Dundee DD2 1SY, UK

**Keywords:** cardiotoxicity, cardio-oncology, heart failure, cancer therapy, precision medicine, pharmacogenetics

## Abstract

Cancer and cardiovascular disease are the leading causes of death in the United Kingdom. Many systemic anticancer treatments are associated with short- and long-term cardiotoxicity. With improving cancer survival and an ageing population, identifying those patients at the greatest risk of cardiotoxicity from their cancer treatment is becoming a research priority and has led to a new subspecialty: cardio-oncology. In this concise review article, we discuss cardiotoxicity and systemic anticancer therapy, with a focus on chemotherapy. We also discuss the challenge of identifying those at risk and the role of precision medicine as we strive for a personalised approach to this clinical scenario.

## 1. Introduction

Survival for patients with cancer has increased in recent decades; doubling in the United Kingdom (UK) in the past forty years [1]. This improvement has, in part, been driven by the development of novel systemic anticancer therapies (including immune checkpoint inhibitors (ICIs) and targeted therapies) and a focus on biomarker-directed therapy [2].

Despite this improvement in outcomes, cancer, alongside cardiovascular disease, remains one of the leading causes of death in the UK [3]. With an ageing adult cancer population [4], who often have features of frailty and several comorbidities [5], the interaction between systemic anticancer therapy (SACT) and cardiovascular disease is an important consideration. Additionally, for those clinicians treating children and adolescents with cancer, reducing the risk of long-term cardiovascular consequences is vital.

The backbone of systemic management for many cancers, both in the paediatric, adolescent, and adult populations, remains chemotherapy [6]. Several chemotherapy agents, in particular anthracyclines, as well as some novel therapies, are associated with acute and delayed cardiotoxicity [6,7]. This can result in significant morbidity and mortality. For example, in the setting of paediatric cancers, survivors are 15× and 7× more likely to develop heart failure and die of cardiovascular disease, respectively, than their noncancer peers [8,9]. In those who develop cardiac dysfunction, prognosis is poor; a 24% 10-year mortality [10]. This has led to the development of a new field of interest in cardiology, cardio-oncology, which focuses on the detection, monitoring, and treatment of cardiovascular diseases occurring as a result of chemotherapy as well as radiotherapy [11].

At present, clinicians treating adult patients with cancer are unable to accurately identify those at an increased risk of chemotherapy-induced cardiotoxicity and cardiovascular complications. This can result in undertreatment or the premature cessation of treatment, particularly in older adults [12]. The challenge is underlined by the lack of consensus on what constitutes a diagnosis of cardiotoxicity, the heterogeneity of the treated population in clinical trials, and the absence of information on older adults [13].

Shared decision making with a patient is essential; the use of baseline left ventricular (LV) function and electrocardiograms, and the clinician’s assessment of age and frailty, coupled to patient discussion, are the main drivers of treatment decisions [14]. The identification of a biomarker for those who are at risk of developing treatment-induced cardiotoxicity would enable a precision-medicine (utilising an individual’s genetics, demographics, or tissue to prevent, diagnose, or treat) personalised approach to therapy—not only in terms of immediate treatment, but also in the approach to longitudinal monitoring and management.

In this concise review article, we will review treatment-induced cardiotoxicity in adult patients with cancer, with a focus on chemotherapeutics. We will also discuss the potential future role of precision medicine in identifying those at risk of developing cardiotoxicity.

## 2. Chemotherapy-Induced Cardiotoxicity

Several definitions of cardiotoxicity exist [15]. The most commonly referenced definition of cardiotoxicity is reduced LV function or failure [16]. However, cardiotoxicity in relation to anticancer therapy can also be broadly defined as any damage inflicted on the heart (functional or structural) from cancer treatments, including SACT and radiotherapy [17]. This can include a spectrum of conditions, including cardiac dysfunction, and effects beyond dysfunction, such as arrhythmias, hypertension, and thromboembolic events. Damage can be thought of as reversible or irreversible [18], although this classification has limitations, as certain anticancer therapies can cause both. The exact scale of this issue is not accurately documented; however, it is estimated that approximately 50% of patients may develop related cardiovascular events up to twenty years after initial SACT treatment [19].

Several classes of chemotherapy are associated with an increased risk of cardiotoxicity (Table 1). These include anthracyclines, antimetabolites, alkylating agents, vinca alkaloids, and taxanes. Each class of chemotherapy has a different mechanism of action (Figure 1) and, as such, produces a variety of cardiotoxic effects.

### 2.1. Cardiac Dysfunction

Cardiac dysfunction is defined by the Cardiac Review and Evaluation Committee supervising trastuzumab clinical trials [26] as: “(1) A decrease in the cardiac left ventricular ejection fraction (LVEF) that was either global or more severe in the septum; (2) Symptoms of congestive heart failure (CHF); (3) Associated signs of CHF, including, but not limited to S3 gallop, tachycardia, or both; and (4) A decline in the LVEF of at least 5% to less than 55%, with accompanying signs or symptoms of CHF, or a decline in the LVEF of at least 10% to below 55%, without accompanying signs or symptoms”.

Several risk factors predispose to cardiac dysfunction [27,28]. These include age, pre-existing cardiovascular disease, combination anticancer therapy (in particular with anti-HER2 targeted therapy, such as trastuzumab), prior cardiac exposure to irradiation, the female sex, and genetics. For older adults, age-related cardiac deconditioning, comorbidities, ischaemia, polypharmacy, and cardiac cell senescence will also all contribute to risk.

Anthracyclines are the most recognised group of chemotherapy agents to cause cardiac dysfunction—with a 5× greater risk than non-anthracyclines [20]. They are used across a range of tumour histologies, settings, and age of patient. Anthracycline agents include doxorubicin (Adriamycin), daunorubicin, epirubicin, and idarubicin. Their anticancer mechanisms of action involve the inhibition of DNA and RNA synthesis, inducing cell membrane damage, the inhibition of topoisomerase II, and the generation of iron-mediated free oxygen radicals [29].

During this process, and their metabolism via mitochondrial NADH dehydrogenase, they also cause injury to cardiomyocytes via lipid peroxidation and the generation of reactive oxygen species (ROS) [30]. The formation of anthracycline-iron complexes may also catalyse a Fenton reaction, further increasing ROS production [31]. Together, these mechanisms induce DNA damage with resultant apoptosis [32]. Importantly, the cardiomyocyte dependence on oxidative metabolism may explain their greater susceptibility to anthracyclines.

Anthracyclines result in irreversible cardiac damage; however, because of cardiac reserves and compensatory mechanisms, e.g., myocyte hypertrophy, the damage often does not become clinically apparent for months or years following administration [33]. The impact is age- and dose-dependent, with children and those aged >65 years at increased risk, and the risk rising exponentially above 300 mg/m^2^ of doxorubicin, or 550 mg/m^2^ of epirubicin [34] (Table 2).

### 2.2. Other Cardiac Implications

An additional group of cardiotoxicities can include arrhythmia, ischaemia, and hypertension. The heart rhythm is reliant on the electrolyte balance, which can be disrupted temporarily, either because of the administration of the chemotherapy agents themselves, or because of subsequent systemic effects, for example, the platinum-induced disruption of renal electrolyte excretion [39]. Indirectly, the side effects of chemotherapy, such as vomiting and diarrhea, can also induce reversible electrolyte imbalance. This, coupled with the use of certain antiemetics, can increase the risk of cardiac rhythm changes, such as QT-prolongation. Rhythm changes can also be caused by structural changes to the heart, such as those induced by anthracyclines [40].

One of the most commonly used groups of chemotherapy agents in solid tumours are the antimetabolites, such as capecitabine and 5-fluorouracil. Both of these agents are associated with coronary artery vasospasm, which presents as atypical chest pain, angina upon exertion or rest, and acute coronary syndromes, including myocardial infarction [21]. Despite underlying cardiovascular disease being associated with an increased risk, this phenomenon is most commonly observed in those with normal coronary arteries, with an incidence between 1–18% [41].

Patients with cancer are at an increased risk of thrombus [42], both venous and arterial. This hypercoagulable state is further increased by the administration of chemotherapy [43]. The best example of this is cisplatin, which has been reported to have a thromboembolic incidence rate between 6.6–18.1% [44,45]. Although the majority of these thrombotic events are peripheral and venous, pulmonary emboli are common and can result in increased pulmonary pressures and right-sided heart failure. In addition, increased arterial embolic events can result in myocardial infarction as well as systemic hypertension.

## 3. Other Anticancer Therapies

As the understanding of the biology of cancer has developed, specific drivers of tumour growth have been identified. Several of these drivers can now be targeted by drugs in clinical use with significant improvements in disease control. An example is the monoclonal antibody, trastuzumab (Herceptin), in HER2-amplified breast and gastric cancer. Furthermore, in the past decade, the treatment paradigm of many cancers has been altered by the emergence of ICIs, which utilise the host immune system to produce an anti-tumoural effect. Both classes of drugs can have off-target cardiac toxicity, which is often challenging to detect, and both have different management approaches.

This is an emerging clinical challenge as, in several tumour groups, there are now multiple combination regimens, including chemotherapy, targeted therapy, and an ICI, making the monitoring and selection of the appropriate management of cardiotoxicity difficult [46].

### 3.1. Targeted Therapies

The most wellrecognised targeted therapy is trastuzumab (Herceptin). This is an IgG1 monoclonal antibody that binds to the extracellular domain of the HER2/ERBB2 receptor, thus inhibiting dimerization and preventing the downstream intracellular signalling pathways. HER2 is amplified in approximately 15–30% of breast cancers, and in 10–30% of gastroesophageal cancers, and these are the main tumour groups in which anti-HER2 therapy is used [47].

Cardiac tissue is also known to express HER2 receptors, which have a protective function in the cardiomyocyte. Trastuzumab reversibly inhibits this function, with subsequent systolic impairment [48]. The rates of systolic dysfunction for single-agent trastuzumab are 3–7% [49], with a higher incidence in combination regimes; for example, in anthracycline-trastuzumab regimes, an incidence of up to 36% has been reported [50]. This observation is supported by preclinical work, which demonstrates increased sensitivity to the anthracycline cardiotoxic effects on the interruption of HER2 signalling [51]. Several other HER2-targeting therapies are in clinical use, including pertuzumab, lapatinib, neratinib, and trastuzumab emastine (TDM-1), with neratinib the only one that does not appear to cause cardiotoxicity.

Other selected targeted agents associated with cardiotoxicity include bevacizumab (used primarily in ovarian and colorectal cancers) and sunitinib (used in renal cell cancer). Bevacizumab is a humanised monoclonal antibody that binds to the ligand of vascular endothelial growth factor A (VEGF-A), inhibiting its binding to endothelial cells. It is primarily associated with hypertension, with an incidence of 4–35% [52]. This hypertension is believed to drive the 2–4% incidence of congestive heart failure [53]. Bevacizumab is also associated with arterial thromboembolic events, including myocardial infarction [54]. This is likely due to a reduction in the anti-inflammatory effects of chronic VEGF exposure, the inhibition of endothelial repair, and a reduction in the levels of nitric oxide and prostacyclin [52].

Sunitinib is a pan-tyrosine kinase inhibitor that binds to several receptors, including VEGF receptors 1–3; stem-cell-factor receptor (c-KIT); the platelet-derived growth factor receptors (PDGFR), alpha and beta; colony-stimulating-factor 1 receptor; FLT3 kinase; and RET kinase. The cardiotoxicity (hypertension and/or LVEF decline) induced by the inhibition of these receptors is an off-target effect, with an incidence of up to 15% [55]. VEGFR inhibition reduces the heart-capillary density and KIT plays a role in the endothelial cell mobilisation to sites of myocardial injury [56]. Patients on tyrosine kinase inhibitors can remain on therapy until disease progression, which can be for several years [57,58]. As such, close monitoring for cardiac toxicity is an important part of the clinical assessment and pathway.

### 3.2. Immune-Checkpoint Inhibitors

ICIs have altered the treatment paradigm of several tumour groups in recent years and are now licenced for a range of histological subtypes across several settings [59,60,61]. However, similar to other antineoplastic agents, they are associated with a risk of cardiotoxicity—albeit with a distinct mechanism from chemotherapy and targeted therapies [62].

The most prominent ICI targets are the host immune suppression proteins, cytotoxic-T-lymphocyte-associated antigen 4 (CTLA4), and programmed cell death 1 (PD-1), along with its ligand, PD-L1. As such, the driving mechanism of the cardiotoxicity of these agents is primarily immune-mediated. The rates of immune-mediated cardiotoxicity (including myocarditis, pericardial disease, conduction abnormalities, and cardiomyopathy) are estimated between 0.1 and 3.2% [63,64] and appear similar for each individual ICI. However, combination therapy is associated with a higher risk [62,65].

Human myocytes have been shown to express both PD-L1 and PD-1 [66], and mice deficient in CTLA-4 and PD-1 are at increased risk of acute myocarditis [67]. There is, therefore, mechanistic support underlying the immune-mediated cardiotoxicity observed. Early identification and treatment with high-dose steroids are essential to preventing arrhythmia, long-term cardiac damage, or death [64,68]. This, however, is made more challenging as the cardiac parameters, including troponin, are not often routinely evaluated in oncology real-world clinical or trial settings. As such, the aforementioned incidence rates may be a significant underestimate of the true rates.

At present, there is no way to identify those at risk. However, it is recognised that patients receiving combination therapy (anti-CTLA4 with anti-PD1/anti-PD-L1), or those with a pre-existing autoimmune disease, are at an increased risk of further immune-related events [69].

## 4. Management of Patients Receiving Cardiotoxic Agents

Both the European Society of Cardiology (ESC) and the European Society of Medical Oncology (ESMO) have produced guidelines for the management of patients receiving a potentially cardiotoxic agent [13,27]. Before the initiation of a potentially cardiotoxic agent, all patients should undergo a thorough cardiovascular assessment, with a specific focus on a history suggestive of ischaemic disease or impaired cardiac function. Blood pressure should be measured and, in those patients receiving multitargeted agents, this, as well as any comorbidities, should be robustly managed before and during therapy. Baseline investigations should include an electrocardiogram, an assessment of the left ventricular ejection fraction, and a measurement of the cardiac biomarkers (troponin and BNP, or NT ProBNP). Consideration should be given to performing a 6-min walk test.

For patients deemed to be at high risk of cardiotoxicity, a prophylactic cardioprotective regime should be initiated: an angiotensin-converting enzyme inhibitor (ACEi)/angiotensin II receptor blocker (ARB) and a beta-blocker [70]. For those receiving an anthracycline, the use of prophylactic dexrazoxane has been shown to reduce the rates of heart failure [71]. Patients with pre-existing cardiac dysfunction should be referred for specialist cardiology review before the initiation of therapy. Careful consideration should be taken of the risk–benefit balance of the treatment and, where possible, modifications to the regime should be made.

While on treatment, for asymptomatic patients, the functional status and cardiac imaging (echocardiogram or MUGA scan) should be performed every 3 months. In those who become symptomatic, imaging should be performed to assess the left ventricular function and the anticancer agent should be interrupted while the investigation is performed.

For low-risk patients who develop asymptomatic or symptomatic cardiac dysfunction, management is dependent on the causative agent. Broadly, targeted therapy and ICI-mediated dysfunction are reversible upon interruption of the drug. In the acute setting, immune-mediated cardiac toxicity (e.g., myocarditis/pericarditis) should be managed with steroids [72]. For those with more prolonged dysfunction, the use of an ACEi/ARB alongside a beta-blocker is recommended [73]. The outcomes are similar to the general heart failure population, and combination therapy appears more effective than the use of ACEi/ARB or a beta-blocker alone [73,74]. There is no role at present for ACEi in the prevention of the development of cardiac dysfunction [27].

Following the completion of therapy, long-term surveillance should be considered for all patients. This is particularly important for those patients who developed cardiotoxicity and are receiving cardioprotective medication. For those who have received anthracyclines, lifelong surveillance should be undertaken, as is recommended for survivors of childhood cancer [75].

## 5. The Role of Precision Medicine

Although several risk factors have been identified, the mechanisms and predictors of chemotherapy-induced cardiotoxicity are poorly understood. It is recognised that early diagnosis and the treatment of cardiotoxicity improves outcomes [73,74]. There are currently no accurate and validated biomarker models to predict which patients are at the highest risk. We are, therefore, unable to prevent cardiotoxicity from occurring or personalise treatment at a precision level. This issue is compounded by the lack of a unified definition of “cardiotoxicity” [76]. The majority of current decisions are made on heuristic learning integrating available biomarkers, including patient assessment, planned treatment, patient samples (e.g., serum), and imaging results.

The identification of a future predictive biomarker will likely necessitate a consideration of the above, as well as the integration of genomic studies (Figure 2). This will ultimately require a translational approach from bench to bedside, incorporating patient samples (including serum and tissue) and pharmacogenomics, with longitudinal clinical outcomes and cardiac imaging data.

At present, a risk assessment requires that a thorough baseline individualised patient history and an examination be undertaken, and the risk factors for cardiovascular disease (and, thus, treatment-induced cardiotoxicity), such as obesity, diabetes, hypertension, and prior cancer treatment, are to be identified, and managed or modified. Clinicians also require knowledge of the baseline cardiac rhythm and function, coupled to the baseline serum markers, such as troponin. Importantly, early changes in troponin have been shown to predict cardiotoxicity in patients with breast cancer and treated with an anthracycline [77]. Cardiac function is assessed primarily using an echocardiogram (ECHO), as per ASCO guidelines [78]. Baseline LVEF does not appear to independently predict who will develop cardiotoxicity [79]. However, the exclusion of patients with reduced LVEF from clinical trials means that there is a reluctance among clinicians to prescribe potentially cardiotoxic regimes to this patient group. The emerging research interest is the potential predictive ability of global longitudinal strain (GLS) and global circumferential strain (GCS), both of which appear to identify those at the greatest risk for the development of cardiotoxicity [80,81].

Together, these factors form a biomarker-based risk stratification for the patient and clinician and will influence the intended anticancer regime and dosing schedule. In the future, patient genetics may also be incorporated, and this influence is a growing current research interest. Genetic changes, in particular, single nucleotide polymorphisms (SNPs) that influence drug absorption, intracellular transport, metabolism, and elimination may play an important role [82]. Recently, genome-wide association studies (GWAS), using a top-down approach, have identified novel genetic variants and their related genes, which are statistically significantly related to chemotherapy-induced cardiotoxicity [83]. Several meta-analyses have been performed. A meta-analysis of 41 studies, including 9183 patients, detected a significantly increased risk associated with six variants: CYBA rs4673, RAC2 rs13058338, CYP3A5 rs776746, ABCC1 rs45511401, ABCC2 rs8187710, and HER2-Ile655Val rs1136201 [83], while another exploring 40 studies (35 using a candidate gene approach and 5 GWAS), with 10,320 patients, identified CELF rs1786814, RARG rs2229774, SLC28A3 rs7853758, and UGT1A6 rs17863783 [84].

Another reported candidate is the SNP, rs246221, encoding a heterozygous carrier of the ABCC1 gene, which plays a role in cellular drug efflux. This SNP was associated with a reduction in the LVEF of >10%, compared to homozygous carriers in patients treated with epirubicin for early breast cancer [85].

Despite the identification of the aforementioned potential candidate genetic changes, the data remains inconsistent and the targets require validation. As such, larger genetic polymorphism studies are required to identify and further elucidate the potential mechanisms of cardiotoxicity.

In Dundee, we have access to a large biorepository, including the Genetics of the Scottish Health Research Register (GoSHARE), and the Genetics of Diabetes Audit and Research (GoDARTS) [86]. These resources enable access to over 200,000 individual patient blood samples, in addition to paired SACT and longitudinal LVEF data, among other variables. This provides a unique platform to investigate the role of specific baseline biomarkers on long-term cardiovascular outcomes. This will be a focus of ongoing research in our unit.

In addition to retrospective genetic analysis, the advent of human-induced pluripotent stem cell models will provide an in vitro system to explore the effect of genomic findings. This will drive hypotheses relating to the underlying mechanisms driving cardiotoxicity and will facilitate the development and investigation of novel strategies to mitigate cardiovascular toxicity [87,88].

## 6. Conclusions

Systemic anticancer therapy is associated with cardiotoxicity through various mechanisms. This has traditionally been associated with chemotherapy, but novel therapies can also impact cardiac function. As the adult cancer population increases in age and, therefore, the predisposition to cardiovascular disease, the effect of SACT on the heart, both in the short- and long-term, takes on more importance. Accurately identifying those who will develop cardiotoxicity is not currently possible. To address this need, a translational bench-to-bedside approach is essential.

## Figures and Tables

**Figure 1 diseases-09-00090-f001:**
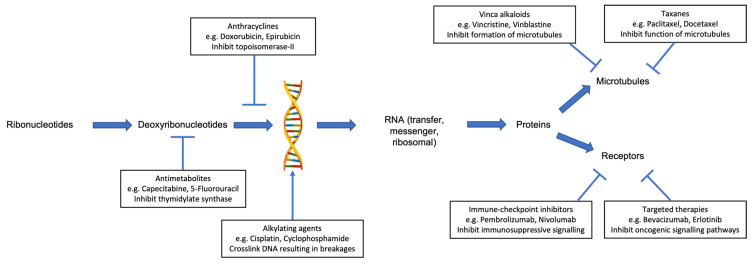
Mechanism of anticancer activity for selected individual classes of systemic anticancer therapy, which are associated with cardiotoxicity.

**Figure 2 diseases-09-00090-f002:**
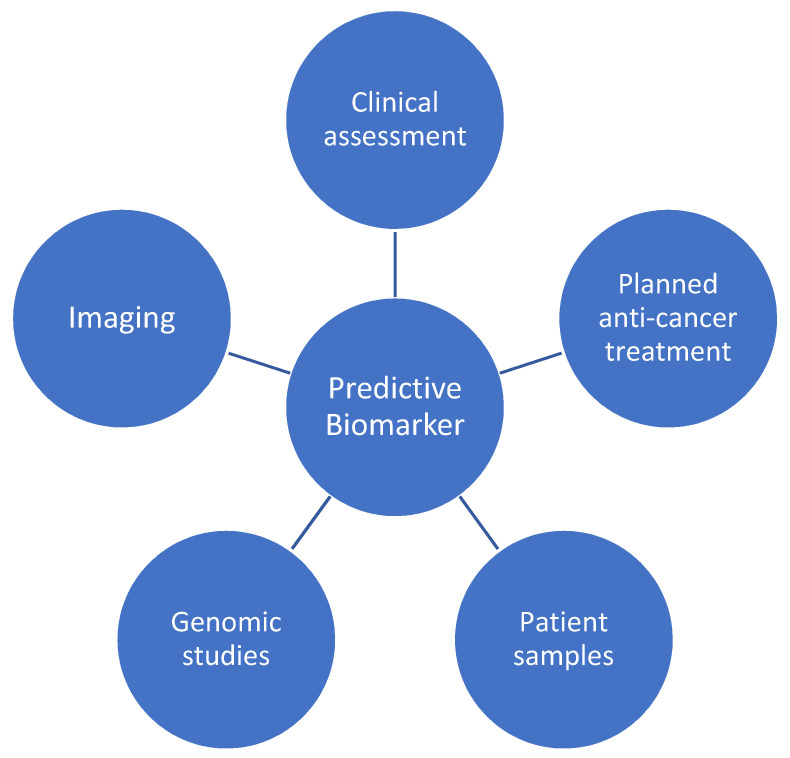
Contributing factors to a future predictive biomarker for anticancer therapy cardiotoxicity.

**Table 1 diseases-09-00090-t001:** Selected chemotherapy agents according to the class of drug and their associated cardiotoxic effects.

Class	Example Drug	Cardiotoxic Effect
Anthracycline	Doxorubicin Epirubicin	Impaired LV function due to irreversible damage to cardiomyocytes [20].
Antimetabolite	Capecitabine 5-fluorouracil	Arterial vasospasm, myocardial ischaemia, and thrombosis [21].
Alkylating agent	Cisplatin	Thrombosis, arterial hypertension [22].
	Cyclophosphamide	Myocardial ischaemia, endothelial cell injury [23].
Vinca alkaloid	Vincristine Vinblastine	Myocardial ischaemia, arterial hypertension [24].
Taxane	Paclitaxel Docetaxel	Myocardial ischaemia, QT prolongation, and bradycardia [25].

**Table 2 diseases-09-00090-t002:** Incidence of left ventricular (LV) dysfunction associated with selected chemotherapy agents.

Chemotherapy Agent	Incidence of LV Dysfunction
Anthracyclines	
Doxorubicin [34]	
400 mg/m^2^	3–5%
550 mg/m^2^	7–26%
700 mg/m^2^	18–48%
Epirubicin [35]	
>900 mg/m^2^	0.9–11.4%
Alkylating agents	
Cyclophosphamide [36]	7–28%
Ifosfamide [37]	
<10 g/m^2^	0.5%
10–18 g/m^2^	17%
Antimicrotubule agents	
Docetaxel [38]	2.3–8%
Paclitaxel [25]	1–10%

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
