# Peer review of "Cardiotoxicity and Chemotherapy—The Role of Precision Medicine"

_diseases, 2021, doi:10.3390/diseases9040090_

Round 1

Reviewer 1 Report

This is a new concept about cardiotoxicity and chemotherapy.

 It is qualified for published.     

 However, Line 56, 61, 71, 91. cancer may be corrected to cancers.

                 Line 157  diarrhoea may be corrected to diarrhea.      

Author Response

Thank-you for reviewing and commenting on our manuscript. Taking all the reviewers comments on board we have provided a revised version, which we hope you will agree to review again.

Reviewer 2 Report

The Study by Viswanathan et al. Urinary 20-HETE: Cardiotoxicity and chemotherapy – the role of precision medicine. But some changes need to be made to the manuscript.

Table 1 shows the chemotherapeutic agents selected according to the drug class, but I think it could be referenced with some studies for the type of drug.
In the introduction there are sentences without the reference. Need to add references.
The findings in the work could be schematized and the review presented could be used in this way.
Diagrams/figures could be mentioned to understand the anti-cancer therapeutic process
The highlights of the work must be placed at the end of the text as well as the limitations of the study.
When it comes to precision medicine, a subject that has great importance in medicine, it would need to be addressed with some articles in the literature to better illustrate and thus generate greater discussion on the subject.

Author Response

Thank-you for taking the time to review our invited concise review manuscript on precision medicine relating to cardiotoxicity and cancer treatment. We feel your comments have significantly improved the article and have detailed how we have addressed them below.

The Study by Viswanathan et al. Urinary 20-HETE: Cardiotoxicity and chemotherapy – the role of precision medicine. But some changes need to be made to the manuscript.

Table 1 shows the chemotherapeutic agents selected according to the drug class, but I think it could be referenced with some studies for the type of drug.

We have now provided appropriate references in Table 1.

In the introduction there are sentences without the reference. Need to add references.

Thank-you for noting this. We have now addressed this.

The findings in the work could be schematized and the review presented could be used in this way.
Diagrams/figures could be mentioned to understand the anti-cancer therapeutic process

We have now added 2 figures relating to the anti-cancer therapeutic process and the potential contributing factors to identify a predictive biomarker.

The highlights of the work must be placed at the end of the text as well as the limitations of the study.
When it comes to precision medicine, a subject that has great importance in medicine, it would need to be addressed with some articles in the literature to better illustrate and thus generate greater discussion on the subject.

We agree with this comment. Therefore, within the limits of the word count for this concise review, we have expanded on our discussion in the hope of addressing your concern and generating greater discussion. The journal requirements for the format of our review article do not require highlights and limitations - as such we have not provided these as yet. If the editor deems them necessary, we would be happy to do this. 

Reviewer 3 Report

Overall Summary: The current review article by Thyla et al accentuate on the role of precision medicine in chemotherapy induced cardiotoxicity. While going through this article, as a reader, I was expecting the author’s discussion on interesting account of recent studies and/or clinical trials conducted in the field of chemotherapy induced cardiotoxicity (Preventive or mechanistic aspects), as there are already several review articles available with similar context (1-13, and many more which I have not mentioned here). This compiled review on the role of precision medicine is not different from already published review articles in the same topic and not adding any new insights. Moreover, the authors have not cited any of these published review articles indicating their negligence in performing review of literature reflecting as a major drawback of this review article. Conducting a good review of literature could have been impacted in a major redeployment of the current article by discussing unsolved puzzles which the previously published review articles missed to mention. Overall, as per my opinion as a reviewer and a reader, in the current format this article has no scope for publication.   

  1. Han X, Zhou Y, Liu W. Precision cardio-oncology: understanding the cardiotoxicity of cancer therapy. NPJ precision oncology. 2017 Sep 12;1(1):1-1.
  2. Polonsky TS, DeCara JM. Risk factors for chemotherapy-related cardiac toxicity. Current opinion in cardiology. 2019 May 1;34(3):283-8.
  3. Linschoten M, Teske AJ, Cramer MJ, Van Der Wall E, Asselbergs FW. Chemotherapy-related cardiac dysfunction: a systematic review of genetic variants modulating individual risk. Circulation: Genomic and Precision Medicine. 2018 Jan;11(1):e001753.
  4. Blaes AH, Thavendiranathan P, Moslehi J. Cardiac toxicities in the era of precision medicine: underlying risk factors, targeted therapies, and cardiac biomarkers. American Society of Clinical Oncology Educational Book. 2018 May 23;38:764-74.
  5. Chung R, Ghosh AK, Banerjee A. Cardiotoxicity: precision medicine with imprecise definitions. Open Heart 5 (2): e000774.
  6. Trapani D, Zagami P, Nicolò E, Pravettoni G, Curigliano G. Management of cardiac toxicity induced by chemotherapy. Journal of Clinical Medicine. 2020 Sep;9(9):2885.
  7. Truong J, Yan AT, Cramarossa G, Chan KK. Chemotherapy-induced cardiotoxicity: detection, prevention, and management. Canadian Journal of Cardiology. 2014 Aug 1; 30 (8): 869-78.
  8. Dessalvi CC, Deidda M, Mele D, Bassareo PP, Esposito R, Santoro C, Lembo M, Galderisi M, Mercuro G. Chemotherapy-induced cardiotoxicity: new insights into mechanisms, monitoring, and prevention. Journal of Cardiovascular Medicine. 2018 Jul 1;19(7):315-23.
  9. Stone JR, Kanneganti R, Abbasi M, Akhtari M. Monitoring for chemotherapy-related cardiotoxicity in the form of left ventricular systolic dysfunction: A review of current recommendations. JCO Oncology Practice. 2021 May;17(5):228-36.
  10. Hahn VS, Lenihan DJ, Ky B. Cancer therapy–induced cardiotoxicity: basic mechanisms and potential cardioprotective therapies. Journal of the American Heart Association. 2014 Apr 22;3(2):e000665.
  11. Angsutararux P, Luanpitpong S, Issaragrisil S. Chemotherapy-induced cardiotoxicity: overview of the roles of oxidative stress. Oxidative medicine and cellular longevity. 2015 Oct;2015.
  12. Avila MS, Siqueira SR, Ferreira SM, Bocchi EA. Prevention and treatment of chemotherapy-induced cardiotoxicity. Methodist DeBakey cardiovascular journal. 2019 Oct;15(4):267.
  13. Florescu M, Cinteza M, Vinereanu D. Chemotherapy-induced cardiotoxicity. Maedica. 2013 Mar;8(1):59.

Author Response

Thank-you for taking the time to review our manuscript and providing your feedback. It has challenged our thinking and we feel has led to a significant improvement in the manuscript.

We recognise that the content was not what you were expecting, however it is important to note that this was an invited review based aimed at providing a broad oversight to general physicians on the topic (rather than specialists in either cardiology or oncology) with a limited word count. We had read all of the review articles that you have kindly provided but in most cases opted to reference the original studies relevant to the section rather than the reviews themselves.

Despite the above, we agree that the discussion on precision medicine required expansion and we have therefore attempted to further expand on this section (again within the limits of the word count). We have also endeavoured to further expand on the other sections of the manuscript and have added two figures.

We hope that our changes and explanation in response to your comments will address your concerns.

Round 2

Reviewer 3 Report

Authors have immensely re-organized the current manuscript with updated review of literature. Additionally, the inclusion of illustrative figures making the article more understandable for readers. Consequently, I recommend the article for publication in Diseases journal. 

Author Response

Thank you once again for your original review comments - they led to a significantly improved manuscript.